# Network pharmacology approach identifies novel anticancer botanicals: Experimental exploration of *Falcaria vulgaris* (Sickleweed) as a therapeutic candidate

Zahra Samadi[1], Eisa Kohan-Baghkheirati[1,2,3*], Madjid Momeni-Moghaddam[1,2], Toktam Hajjar[1,2], Zahra Ghavidel[1]

**1** Department of Biology, Faculty of Science, Hakim Sabzevari University, Sabzevar, Iran, **2** Research Group of Systems Phytopharmacology, Hakim Sabzevari University, Sabzevar, Iran, **3** Non-Communicable Diseases Research Center, Sabzevar University of Medical Sciences, Sabzevar, Iran

\* e.kohan@hsu.ac.ir

## Abstract

Cancer remains a major global health challenge with limited therapeutic options. Medicinal plants have emerged as promising candidates for anticancer drug discovery. This study introduces an innovative network pharmacology approach to predict potential anticancer botanicals by analyzing their metabolite profiles. Through bipartite network analysis of 3,250 plants and 667 experimentally validated anticancer metabolites, we identified 61 top-ranked plants, among which 85.25% exhibited known anticancer properties, while 14.75% represented novel candidates. Further experimental validation of *F. vulgaris* extract revealed its predicted anticancer potential. Biochemical analyses demonstrated high peroxidase activity (0.063 AU/min per mg protein), substantial flavonoid content (178.33 ± 18.3 mg QE/g), and elevated total phenolics (643.3 ± 20.8 mg GAE/g). GC-MS analysis identified 22 bioactive compounds, predominantly featuring documented antioxidant, anticancer, and anti-inflammatory properties. The extract exhibited significant dose-dependent cytotoxicity against breast cancer cell lines, with MTT assays showing significant inhibition ($p < 0.0001$) in both MCF-7 and 4T1 cells. Flow cytometry analysis further confirmed remarkable cell death induction, with rates reaching 90.51% (MCF-7) and 97.4% (4T1) at 10 mg/mL concentration of the extract. These experimental results robustly validate our network pharmacology approach for efficiently identifying anticancer botanicals through their metabolite profiles. This integrative strategy connects computational prediction with experimental validation, providing a scalable framework for discovering plant-derived therapeutics. Our findings not only explore *F. vulgaris* as a promising candidate but also cover the way for systematic exploration of understudied botanicals in oncology.

**Data availability statement:** The data may be found as Supplementary file of the manuscript.

**Funding:** The author(s) received no specific funding for this work.

**Competing interests:** The authors have declared that no competing interests exist.

## Introduction

Cancer, a widespread and pressing global concern, affects millions of individuals annually and stands at the forefront of disease-related deaths particularly in developing nations [1,2]. Cancer originates from diverse cells within the body that early stages may not exhibit specific symptoms, while advanced stages apparent distinct signs [3]. Conventional treatments for cancer include surgery, radiotherapy, chemotherapy, hormone therapy, and targeted therapy. Nevertheless, the effectiveness of these treatments is not always guaranteed and they can contribute to drug resistance or result in adverse side effects [4–6]. Consequently, there has been a rapidly increasing interest in exploring alternative approaches, including the utilization of medicinal plants and their active compounds, which may offer distinct safety or tolerability advantages in specific contexts [7–10].

Targeted therapies (e.g., kinase inhibitors) often face limitations due to single-pathway inhibition, leading to drug resistance and narrow therapeutic windows. In contrast, plant-derived extracts offer multi-target synergies that can simultaneously modulate proliferation, apoptosis, and oxidative stress pathways, potentially overcoming these limitations [7,8]. Growing clinical evidence highlights their potential to improve safety profiles when combined with conventional therapies – for example, herbal interventions have been shown to significantly reduce chemotherapy-associated nausea, fatigue, preventing weight loss, relieves the gastrointestinal toxicity, and hematological toxicity in randomized trials [9,10]. Controlled trials demonstrate plant extracts' superior safety profiles versus targeted therapies as a cross section study on clinical database reveals that plant extracts have less side effect in comparison to synthetic drugs, like Mistletoe (*Viscum album*) extract vs. erlotinib, Turmeric (*Curcuma longo*) extract vs. Bevacizumab, and Ginseng vs. Tyrosine kinase inhibitors [9]. This enhanced tolerability stems from natural buffering by complementary phytochemicals that protect non-target tissues while maintaining anticancer efficacy [11,12]. Additionally, plant-based therapies demonstrate superior cost-effectiveness, with production costs for vinca alkaloids being orders of magnitude lower than synthetic biologics [13]. These advantages position plant extracts as promising complements to existing targeted therapies.

With thousands of plant species discovered to possess medicinal properties, these botanical wonders have shown promise in directly combating diseases, inhibiting disease progression, and alleviating side effects [11–13]. The therapeutic effects of medicinal plants can be attributed to the presence of secondary metabolites, which are organic compounds abundantly found in plants. Among these metabolites, some have been scientifically validated for their potential anticancer properties [14–17]. However, the intricate relationship between the composition of plant metabolites and their specific effects on diseases continued as hard challenge. To address the challenge, this study aims to utilize network analysis approach to identify potential anticancer plants based on their metabolite composition.

Every year, extensive experimental analysis is conducted to evaluate the anticancer properties of plants [18,19]. Creating a well-structured list of potential anticancer plants based on verified anticancer metabolites can significantly reduce the time and cost associated with evaluating plants. This approach avoids unnecessary costs from

testing irrelevant plant species. A ranked list generated by analyzing the biological networks of plant metabolites -using topological features (e.g., node centrality, connectivity, or radiality)- becomes invaluable in such a scenario [20]. Our proposed method can be employed to select the most effective anticancer plants from a list of unverified plants. As a result, plants with higher scores on the list are more likely to possess anticancer properties.

In order to validate the effectiveness of the approach mentioned above, we conducted an examination of the cytotoxic and apoptotic properties of the extract from *F. vulgaris*, with common name of Sickleweed or Longleaf which belongs to the Apiaceae family, on Breast cancer cell lines as model. Standard breast cancer therapies include endocrine agents for hormone receptor-positive tumors, HER2-targeted agents, and cytotoxic chemotherapy for triple-negative breast cancer (TNBC), and emerging targeted therapies (CDK4/6 inhibitors, PARP inhibitors, immunotherapy) [4]. However, treatment resistance and toxicity limitations persist, motivating the search for novel phytochemical alternatives with distinct mechanisms of action.

Based on traditional medicine and experimental investigations, the recognized therapeutical properties of *F. vulgaris* includes antioxidant, antimicrobial, anti-inflammatory, anti-diabetic, and bleeding-inhibiting [21,22]. This plant is rich in various compounds, including anthraquinones, alkaloids, phenols, tannins, saponins, steroids, flavonoids, as well as anticancer compounds like caryophyllene oxide and Spathulenol [23–25]. Further analysis using gas chromatography–mass spectrometry on *F. vulgaris* confirmed the presence of key components such as Carvacrol, Spathulenol, and α-pinene [26]. However, there is currently limited evidence on the specific anticancer effects of *F. vulgaris*. This study opens new approach to get new medicinal plants for cancer treatment.

## Materials and methods

### Anticancer metabolite list

To obtain a comprehensive list of herbal anticancer metabolites, we compiled experimentally validated metabolites through systematic searches of PubChem and KNApSAcK [27] using keywords ("anticancer", "anti-tumor", "cytotoxic") with three inclusion criteria: (1) direct experimental evidence in peer-reviewed studies, (2) plant origin confirmation, and (3) reported mechanisms of action. The KNApSAcK_Family database was utilized to retrieve plants containing listed metabolites [27]. An unweighted bipartite network was constructed in Cytoscape 3.9.1, where edges represent metabolite-plant associations without predefined weighting. Node categories included: (i) plant species (annotated with taxonomic names) and (ii) metabolites (annotated with KNApSAcK IDs).

### Network analysis

To find key plants in the network as potential anticancer plants, network was analyzed using 8 topological parameters of Cytoscape's Cytohubba and Network analyzer extensions (Degree, Betweenness, Closeness, Edge percolated component, Eccentricity, Radiality, Stress, and Bottleneck scores; For more details like mathematical formulations and calculation algorithms refer to S3 Table). Each parameter generated a ranked list of top 100 nodes, with final plants prioritization based on their frequency across these lists. For validation, we employed: (1) temporal validation by selecting *F. vulgaris* (rank #22, appearing in 6/8 top-100 lists) before any anticancer reports existed (2019 Sep.-2021 Jul.); (2) experimental confirmation via antioxidant (DPPH), cytotoxicity (MTT), and apoptosis (flow cytometry) assays; and (3) retrospective literature validation to concordance between top-ranked plants and known anticancer species. While our unweighted network approach prioritized topological features over bioactivity potency and omitted computational cross-validation, the experimental confirmation supports the predictive value of this methodology.

### Plant extraction

The above-ground parts of *F. vulgaris* were obtained from the local market in Sabzevar (Iran), identified by plant biologist (Herbarium code; HSUH4105), and dried after washing to remove any surface pollution. A hydroalcoholic extract was

prepared by combining 1–10 (w/v) plant material with 70% ethanolic solvent, shacked at 180 rpm for 24 hours at room temperature. The extract was filtered by Whatman paper No.1 and concentrated using a rotary evaporator at 35°C and 50 rpm. The concentrated extract was dried in an oven at 35°C for 24 hours. The final product was stored at −20°C for next experiments.

## Antioxidant assay

To measurement total flavonoids the calorimetric aluminum chloride method was employed [28]. A 5 µg of the extract was diluted with 5 ml distilled water and 300 µl of 5% sodium nitrite was added. After 5 minutes, 600 µl of 10% aluminum chloride was added to the solution. After 6 minutes, solution was mixed with 2 ml of 1M sodium hydroxide and 2 ml of water. The absorption intensity was measured at 510 nm. Flavonoids concentration was calculated based on Rutin mg/100 gFW.

The phenol content of the extract was determined using Folin-Ciocalteus phenol reagent through spectrophotometry [29]. A standard curve was generated using Gallic acid. A 0.5 ml of extract (10 µg/ml) was mixed with 0.5 ml of 10% Folin-Ciocalteu reagent and 1.5 ml distilled water. After 5 minutes, 2 ml of 5% sodium carbonate was added to the mixture. The solution was vortexes and the absorbance was measured at 720 nm [30].

The antioxidant activity was determined using the DPPH method. A 400-ppm plant extract powder was prepared with a methanol solvent and combined with DPPH. Synthetic antioxidants of BHT and Gallic acid in methanol solvent were used for standard curve. The 150 µl of DPPH methanolic solution was mixed with 100 µl plant extract in a 96-well microplate and shaken. The absorbance of the samples was recorded at 517 nm in comparison to the control after 30 minutes incubation in the dark [31,32]. The $IC_{50}$ values for BHT and Gallic acid were calculated and used to express the antioxidant activity of the extract based on below formula;

$$\text{GEAC (mg GA/g dw)} = IC_{50} \text{ (GA)}/ IC_{50} \text{ (sample)} \times 1000$$

To measure the peroxidase activity, 3 ml phosphate buffer (50 mM) was mixed with 7 µl hydrogen peroxide (30%) and 6 µl guaiacol (20%) as the electron donor. The spectrophotometer was calibrated with this mixture. Next, 15 µl of the extract was added to the mixture and the absorbance was read at 470 nm. Enzyme activity was calculated as formation of Tetraguaiacol µl/min in the one gram of extract [33].

## Cell culture, MTT assay and Flow cytometry

This study was performed in line with the ethical policies and procedures approved by Hakim Sabzevari University ethics committee (Approval No. IR.HSU.REC.1399.012). The 4T1 (ATCC CRL-2539) and MCF-7 (ATCC HTB-22) cell lines were purchased from Sabzevar University of Medical Sciences's cell bank to evaluate anticancer effects of *F. vulgaris* extract. MCF-7 and 4T1 cells were cultured in DMEM and DMEM+ RPMI medium respectively. Each medium was contained 10% FBS (Gibco), 1x penicillin and streptomycin antibiotics (Gibco). Cells were grown in a highly regulated environment of 37°C, 5% CO2, and 95% humidity (memmert incubator). To assessed cell viability, 100µl (104 cells) of MCF-7 and 4T1 cells were seeded into each well of 96-well plates and incubated in 5% CO2 air at 37°C. In the next step, the cells were treated with different concentrations of extract of *F. vulgaris* (0 mg/ml as solvent control contains same volume ddH2O as in treated groups, 0.15, 0.31, 0.62, 1.25, 2.5, 5, 10, 20, 30, 40, 50, 60, 70, 80 and 90 mg/ml) for 24, 48, and 72 hours. Subsequently, RPMI medium (100µl) and MTT (10 µL) were added to each well and incubated at 37°C for 4 hours [34,35]. The OD was recorded at 570 nm on a microplate reader (Elisa reader).

$$\% \text{ Cell viability} = ((\text{abs\_sample} - \text{abs\_blank})/(\text{abs\_control} - \text{abs\_blank})) \times 100$$

In addition to our cancer cell line testing, the selectivity of *F. vulgaris* extract was preliminarily evaluated using the 3T3 mouse embryo fibroblast cell line (ATCC CRL-1658) as a model for normal cells. Cells were cultured and maintained in

DMEM supplemented with 10% FBS and 1% penicillin-streptomycin. Subsequently, they were treated with the hydroalcoholic extract at concentrations ranging from 0 to 90 mg/mL for 24 hours. Cell viability was assessed using the MTT assay as mentioned above. The high concentration range was selected based on the crude nature of the extract, which necessitates higher doses to achieve a pharmacological effect due to limited bioavailability of active constituents. This dose selection is further supported by existing literature demonstrating the efficacy and safety of similar high concentrations of *F. vulgaris* extracts both *in vitro* [22] and *in vivo* [23].

Cell death rates were assessed using Annexin V-FITC and PI staining (IQ Products Company, Netherlands) in 4T1 and MCF7 cell populations ($5 \times 10^5$ cell per each well of 6-well plates) that were treated with *F. vulgaris* extract (0 mg/ml as solvent control contains same volume ddH$_2$O as in treated groups, 10 mg/ml, and 30 mg/ml). The cells were resuspended in 100 µl of binding buffer (1X) and then mixed with 5 µl of Annexin V-FITC and 5 µl of PI for 20 minutes in the dark at room temperature. After incubating for 5 min, the samples were analyzed using CYTEK flow cytometry [34]. Data analysis was performed using FLOWJO 7.6.1 software.

### Gas chromatography-mass spectrometry (GC-MS) analysis

The GC-Mass device was used to determine the composition of the extract. The analyses were performed on a GC/MS (Shimadzu GC-MS QP 2010, Shimadzu) equipped with a HP-5MS (length: 30 m; Diameter: 0.25 mm; film thickness: 0.25µm) Capillary column. The injections were performed in the split mode (50.1), and the injector temperature was set to 260°C, while the carrier gas was helium at a flow rate of 1 mL/min. The temperature in the column was held at 60°C for 4 minutes in order to condense the hydrocarbons. The temperature was then increased to 100°C at 3°C/min and held for 2 minutes. After that, the temperature ramped to 250°C at 4°C/min and stayed for 5 minutes. Finally, the components were identified based on Willey 7n.l GC/MS database and spectrums were evaluated by APEX.

### Statistical analysis

Statistical differences between control and treated groups were analyzed using one-way ANOVA and Tukey test. Graph Pad Prism 8 software was used for analysis and visualization of results. All groups have had four replicates and significance level was considered as P value <0.05.

## Results

### Data collection and network analysis

A total of 132000 data points pertaining to anticancer plant metabolites were collected. Data retrieval was conducted from databases, resulting in a total of 667 anticancer metabolites whose effects were confirmed by laboratory data (S1 Table). A comprehensive evaluation using the KNApSAcK_Family database revealed that these metabolites are produced by 3250 different plants. Table 1, presents the anticancer metabolites found in over 100 plants. The quantitative distribution of these anticancer metabolites across plant species is particularly noteworthy, as phytochemical synergy research suggests that plants containing multiple bioactive compounds may offer enhanced therapeutic potential through multi-target effects [14].

Plant-metabolite binaries were visualized as a bipartite network with 3918 nodes and 5392 edges (Fig 1a). The network properties and node degree distribution (number of neighbors per node), show that most nodes in the network have a degree of less than ten. Network analysis revealed that 8 out of 11 parameters of Cytoscape's extensions could distinguish the importance of nodes in the network. As a result, 100 top nodes per parameter (a total of 800 top nodes, where node refers to a plant) were identified. The list of top plants was ranked in descending order based on their replications in the mentioned parameters. Interestingly, the results showed that twelve plants had 8 replications while 107 plants only had one replication (Fig 1d).

**Table 1. Main distribution of anti-cancer metabolites in plant species.** Name, formula/structure, and biological activity of anti-cancer metabolites that were found in more than 100 plants. Degree represents the number of plants that are containing the metabolite. PubChem ID has active link which shows the metabolite's information including structure, properties, toxicity, and so on.

| Metabolite ID | Degree | Name | Formula | Biological activity | PubChem ID |
|---|---|---|---|---|---|
| C00002526 | 244 | Genistein | $C_{15}H_{10}O_5$ | Cytotoxic, Anticancer | 5280961 |
| C00003749 | 204 | Lupenol | $C_{30}H_{50}O$ | Antineoplastic | 259846 |
| C00005373 | 178 | Isoquercetin | $C_{21}H_{20}O_{12}$ | APN inhibitor inactive | 5280804 |
| C00000615 | 159 | Caffeic acid | $C_9H_8O_4$ | Affect DNA binding | 689043 |
| C00003741 | 156 | Betulinic acid | $C_{30}H_{48}O_3$ | Cytotoxic, Anticancer | 64971 |
| C00002647 | 126 | Gallic acid | $C_7H_6O_5$ | Cell growth inhibitor | 370 |
| C00002724 | 123 | Chlorogenic acid | $C_{16}H_{18}O_9$ | Cytotoxic, Anticancer | 1794427 |
| C00005372 | 120 | Hyperin | $C_{21}H_{20}O_{12}$ | APN inhibitor inactive | 5281643 |
| C00001819 | 114 | Berberine | $C_{20}H_{18}NO_4$ | Cytotoxic, antiprolif. | 2353 |
| C00001110 | 109 | Vitexin | $C_{21}H_{20}O_{10}$ | Antithyroid activity | 5280441 |
| C00003740 | 105 | Betulin | $C_{30}H_{50}O_2$ | Antineoplastic | 72326 |

### Identifying potential anticancer plants

To identify the top plants with anticancer properties, we used a cut off of ≥5 replications. As a result, sixty-one plants were included in the list (Fig 1b and S2 Table). A literature review revealed that 85.25% of these plants had previously established anticancer properties, confirming the validity of our methodology. Among those, the top ten plants with the highest replication (i.e., eight) are detailed in Table 2 and S1 Fig. Those plants are from different families and have effectiveness on various types of cancers especially breast cancer. The Venn diagram analysis showed that top anticancer plants have 6–61 known anticancer metabolites (Table 2 and S1 Fig.), which Ferolic Acid (C00002743), 3-O-Caffeoylquinic acid (C00002724), Caffeic Acid (C00000615), Hirstrin (C00005373), Hyperin (C00005372), Naringenin (C00000982) and p-Coumaric Acid (C00000152) have had the highest frequency.

The remaining percentage that was not experimentally analyzed (14.75% - nine plants), could be interestingly considered as potential new anticancer plants (Fig 1c, Table 3 and S2 Fig.). However, further experimental examinations are needed to determine their effectiveness, as we report with *F. vulgaris* in this study. This selection was based on a multi-criteria rationale designed to prioritize a candidate with high translational potential. First, *F. vulgaris* has a well-documented history of traditional use for wound healing and treating gastrointestinal disorders, supported by modern studies confirming its antioxidant, antimicrobial, and anti-inflammatory properties [21,22,36–38]. This established bioactivity profile suggests a favorable safety and pharmacologic potential. Second, phytochemical analyses, including GC-MS, have identified a rich composition of bioactive compounds in *F. vulgaris*—such as flavonoids, caryophyllene oxide, spathulenol, carvacrol, and α-pinene [23–26]—many of which have demonstrated anticancer effects in other systems. Crucially, despite this promising phytochemical foundation, there remained a significant gap in direct experimental evidence for its efficacy against specific cancers. Finally, as a native species, its regional availability aligns with broader goals of investigating locally-sourced, understudied medicinal plants. While other top-ranked candidates (e.g., *Diospyros kaki*, *Sedum takesimense*) also present strong computational profiles and merit future investigation, the combination of traditional evidence, phytochemical abundance, and a clear research gap positioned *F. vulgaris* as an ideal candidate for this proof-of-concept study.

### Antioxidant and anticancer potency of *F. vulgaris*

To assess the quality of the *F. vulgaris* extract, several parameters were measured, including DPPH antioxidant activity, peroxidase activity, total flavonoid content, and phenol content. The DPPH radical scavenging assay (RSA) was used to evaluate the antioxidant activity of *F. vulgaris* extract (Fig 2a), and $IC_{50}$ value of 3.14 mg/ml was obtained, in comparison

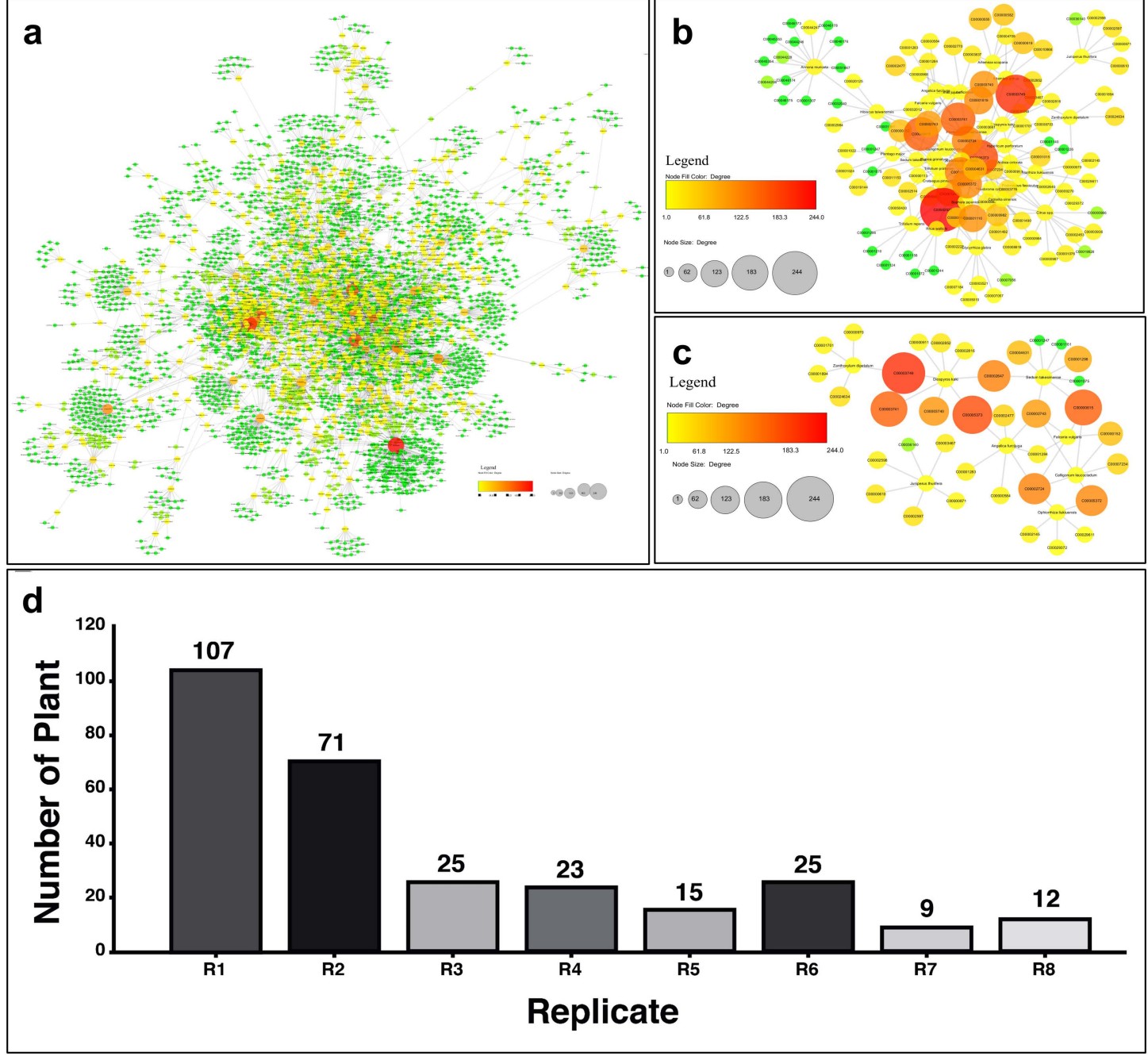

**Fig 1. Plant-metabolites bipartite network characterization.** a: Plant-metabolite network overview with 3918 nodes and 5392 edges, the node's size and color (green to red) is correlated with the node degree (i.e., from 1 [green] to 244 [red]). b-c: The subnetwork of respectively 61 superior plants and 9 potential anticancer plants. d: The number of plants in descending order based on their replications (i.e., how often each plant appeared in top-100 rankings across all network analyzing parameters).

to gallic acid (4.74 mg/ml) and BHT (4.43 mg/ml) as reference compounds, indicating *F. vulgaris* reveals high antioxidant potency. The IC50, representing the extract concentration required to neutralize 50% of free radicals, was determined from dose-response curves of RSA using linear regression analysis of the concentration-RSA relationship.

**Table 2. Top ten documented antic-cancer plants.** The detailed information of top ten plants with the highest replication (i.e., eight) that their anti-cancer properties were previously established.

| Plant Species | Family | AC met* | Cancer type to established effect |
|---|---|---|---|
| *Punica granatum* | Lythraceae | 61 | Breast, colon, prostate |
| *Psidium guajava* | Rutaceae | 26 | Breast, osteosarcomas, leukemia |
| *Salvia officinalis* | Lamiaceae | 22 | Breast, lung, colon |
| *Trifolium pratense* | Fabaceae | 22 | Breast, oral cancer |
| *Phellodendron amurense* | Rutaceae | 14 | Leukemia, lung, duodenum, |
| *Taxus baccata* | Taxaceae | 14 | Breast, cervical, leukemia |
| *Phyllanthus emblica* | Euphorbeaceae | 10 | Breast, lung, liver, cervical, |
| *Plantago major* | Plantaginaceae | 9 | Breast, ovarian |
| *Theobroma cacao* | Malvaceae | 8 | Breast, colon |
| *Viscum coloratum* | Santalaceae | 6 | Uterine carcinoma, breast, gastric |

*Number of known anti-cancer metabolites that were realized by Venn diagram analysis.

**Table 3. The detailed information of nine potential/predicted anti-cancer plants.**

| Plant Species | Family | AC met* | Replication |
|---|---|---|---|
| *Diospyros kaki* | Ebenaceae | 8 | 8 |
| *Angelica furcijuga* | Apiaceae | 7 | 6 |
| *Calligonum leucocladum* | Polygonaceae | 6 | 6 |
| *Juniperus thurifera* | Cupressaceae | 6 | 5 |
| *Sedum takesimense* | Crassulaceae | 5 | 6 |
| *Ophiorrhiza liukiuensis* | Rubiaceae | 5 | 5 |
| *Zanthoxylum dipetalum* | Rutaceae | 5 | 6 |
| *Rhus wallichii* | Anacardiaceae | 3 | 5 |
| *Falcaria vulgaris* | Apiaceae | 6 | 6 |

The level of peroxidase activity was also investigated which shown concentration dependency with 0.063 AU/min per mg of protein at concentration of 60 mg/ml of the extract (Fig 2b). The Folin-Ciocalteu method was employed to measurement the total phenol content which the phenol level was 643.3 ± 20.8 mgGAx/g extract of *F. vulgaris* (Fig 2c–2d). The flavonoid content of the extract was determined using a colorimetric method (Fig 2e–2f) and displayed 178.33 ± 18.3 mgQEx/g extract. The results of antioxidant assay show that *F. vulgaris* has a variable potential antioxidant capacity in comparison to other plants like *Stinging nettle* [39,40].

The MTT assay showed that the toxicity of *F. vulgaris* extract on both MCF-7 and 4T1 breast tumor cell lines were concentration-dependent. While the extract showed no significant growth inhibition (p > 0.05) in normal 3T3 cells at concentrations up to 30 mg/mL after 24 h exposure (S4 Fig.), concentrations above 5 mg/ml of the extract had a significant inhibitory effect (p < 0.0001) on cancer cell lines at different time points of 24, 48 and 72 hours (Fig 3a–3f). However, the lowest percentage of cell survival and the highest lethality were observed at a concentration of 30 mg/ml of the extract. The *F. vulgaris* extract exhibited potent cytotoxicity with IC50 values of 5.32 mg/mL in MCF-7 and 5.99 mg/mL in 4T1 cells (24h MTT assay) (S4 Table).

The flow cytometry results also showed that the cell death rate of the MCF-7 cell line, treated with a 10 mg/ml *F. vulgaris* extract was 90.51% (Fig 4). At a concentration of 30 mg/ml the cell death rate increased to 93.3%. On the other hand, in the 4T1 cell line, a concentration of 10 mg/ml resulted in 97.4% cell death, while at a concentration of 30 mg/ml, 87.2% cell death was observed (Fig 4).

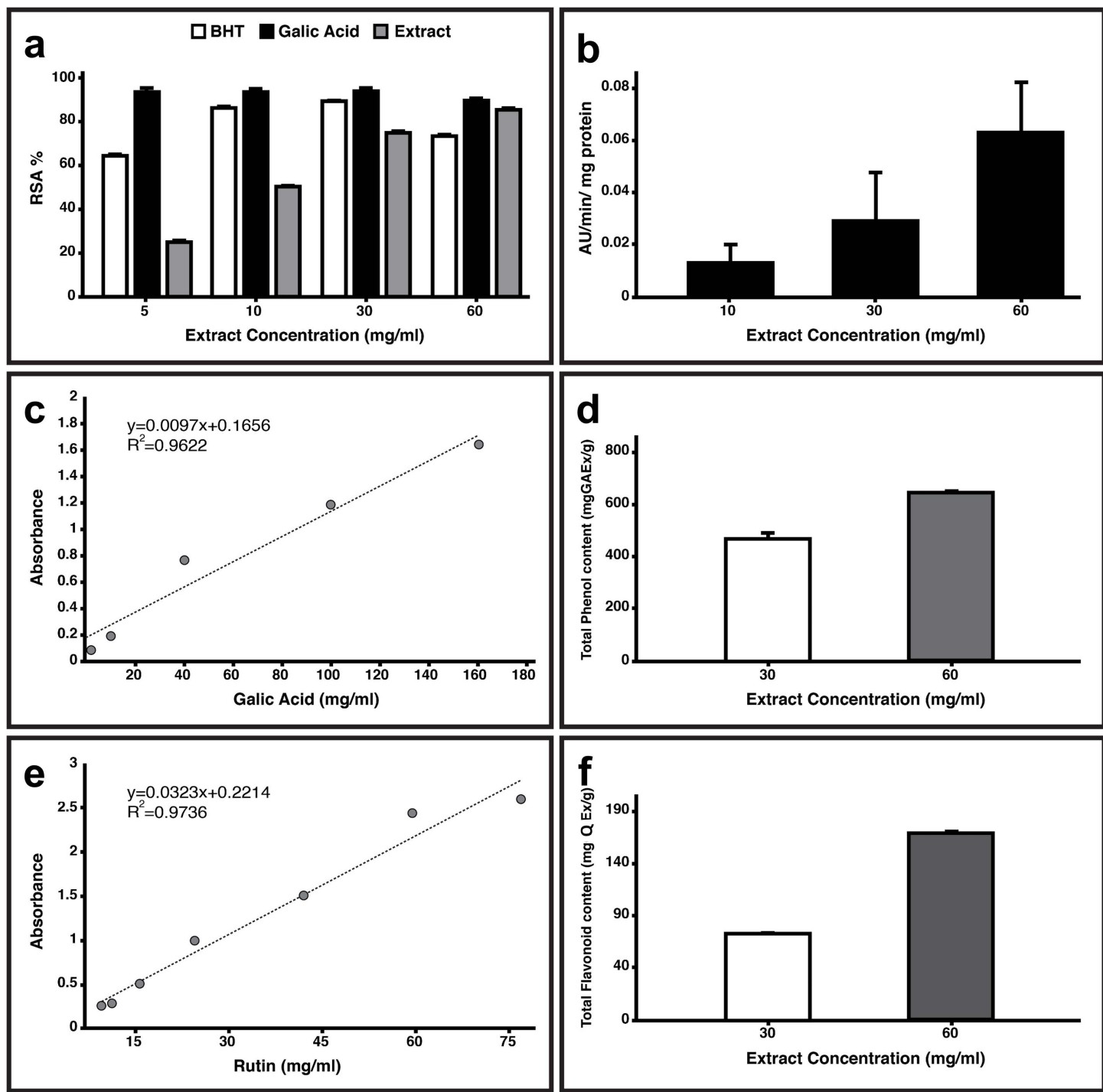

**Fig 2. Evaluation of antioxidant activity of *Falcaria vulgaris*.** a: The DPPH radical scavenging assay (RSA) of *F. vulgaris*. b: The peroxidase activity at three concentration (10, 30, 60 mg/ml) of the extract. c-d: Standard curve of Gallic acid (mg/ml) along with comparison of total phenol (mg GAEx/g) at 30 and 60 mg/ml concentrations of the extract. e-f: Standard curve of Rutin (mg/ml) along with flavonoid (mg QEx/g) content at 30 and 60 mg/ml concentrations of the extract. Data is represented as Mean ± SD, n = 3.

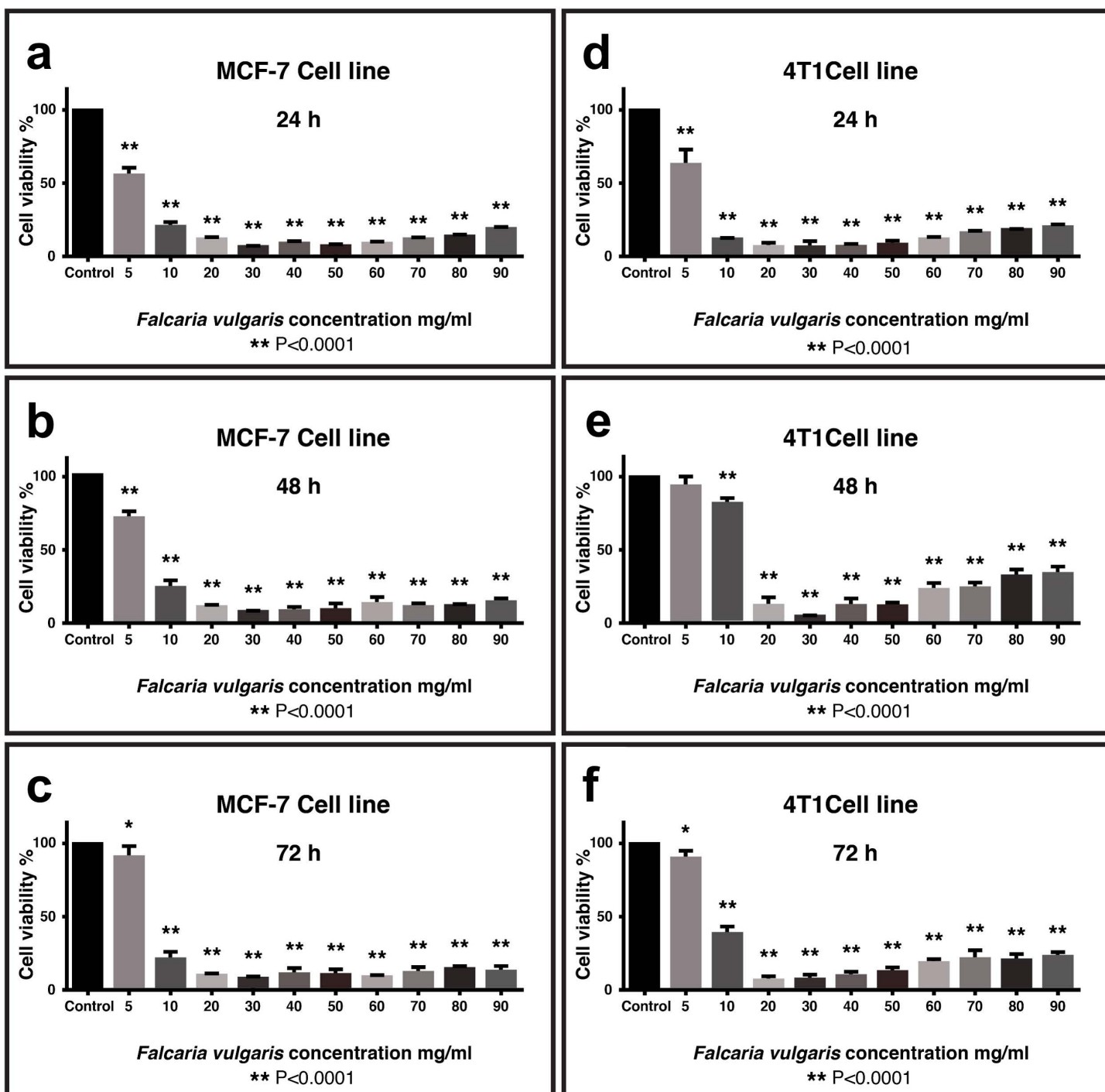

**Fig 3. MTT assay (cytotoxicity) of *F. vulgaris* extract on MCF-7 (a-c) and 4T1 (d-f) breast cancer cell lines at 24, 48, and 72 h after treatment.** Data analyzed by one-way ANOVA with Tukey test (n = 4 replicates; **p < 0.0001 vs control).

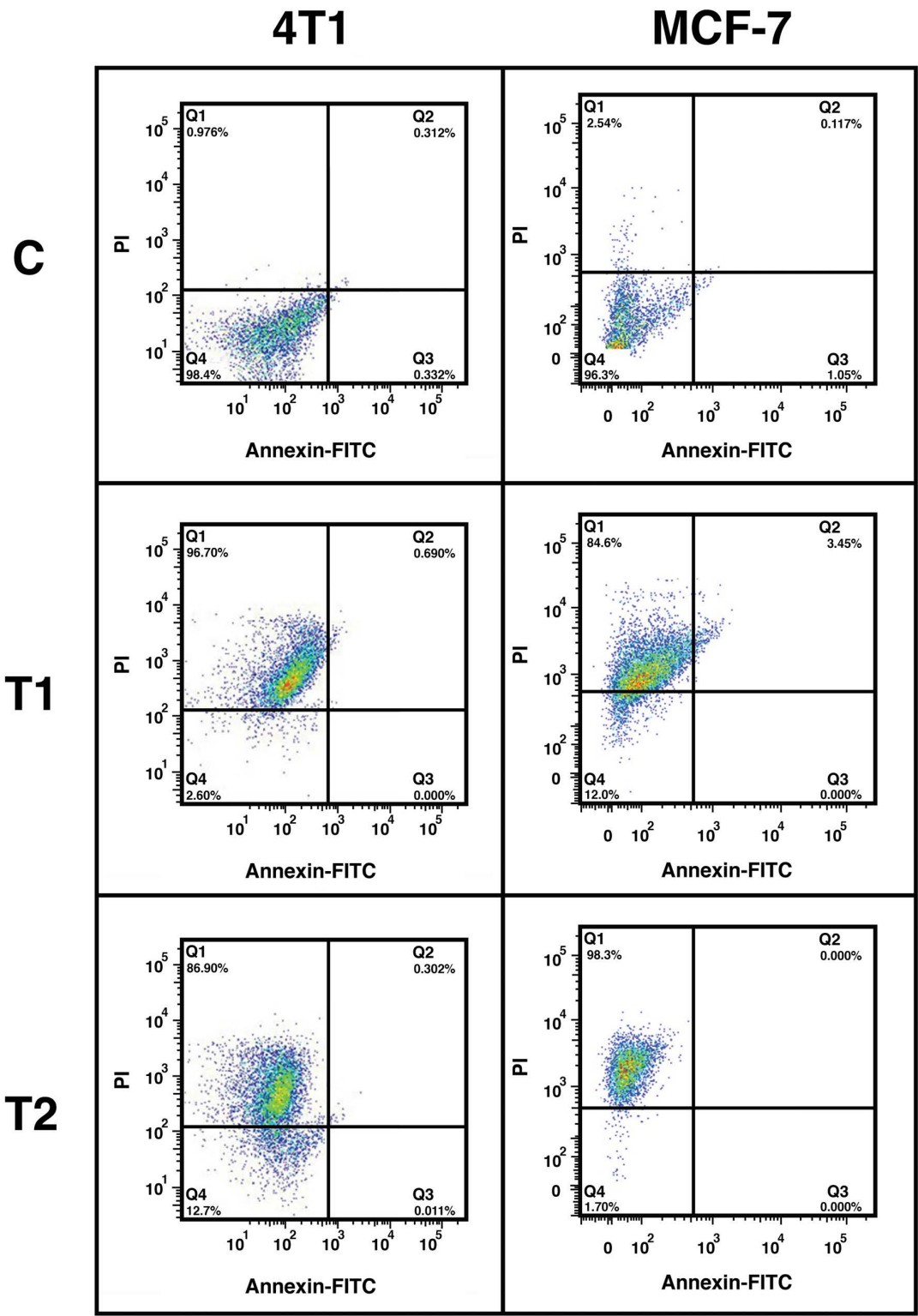

**Fig 4. Flow cytometry (cell death rate) on 4T1 and MCF7 cell populations that were treated with 0 mg/ml (C, as solvent control, same volume ddH$_2$O as in treated groups), 10 mg/ml (T1) and 30 mg/ml (T2) concentrations of *F. vulgaris* extract.** Numbers in each quadrant reflects the percentage of cells (population).

**GC-MS Based metabolite profiling of *F. vulgaris***

The bioactive compounds of *F. vulgaris* hydroalcoholic extract were identified by GC-MS analysis. Table 4 and S3 Fig. show the chromatogram and details like CAS number, retention time (RT), concentration (peak area %), and quality of 22 active compounds in the extract. The dominant metabolites in the plant extract includes various terpenes and terpenoids, alkanes, Steroids and lipids, and phenols which may be responsible for potential therapeutic properties of *F. vulgaris*. About fifty percent (49.56%) of the extract is made up of fatty acids and steroids like Palmitate, Hexadecenoic acid, Linoleate, and Octadecadienoic acid which their individual therapeutic roles were investigated. The major second group of metabolites (more than 30 percent of peak area) in *F. vulgaris* extract are terpenes/terpenoids including Spathulenol, Loliolide, Pentadecanine, Phytol, Stigmasterol and Gama-sitosterol that commonly known with their anticancer, antimicrobial, anti-inflammatory, antioxidant, and antiallergic properties. Some alkanes including Neophytadiene and Eicosane were also observed in the extract.

## Discussion

Data mining and network analysis were utilized in this study to identify potential anticancer plants based on their metabolites. The network consisting of 667 anticancer metabolites and their respective host plants was carefully examined. The use of bipartite networks, alongside the Cytohubba extension, proved to be highly effective in identifying significant and central nodes, also known as hubs (herein, plants). This approach demonstrated remarkable efficiency in determining the most important plants within the network. Consequently, a total of sixty-one anticancer plants were identified, and a literature review confirmed that 85.25 percent of these plants had scientifically established anticancer properties. This

**Table 4. Quantification results of *F. vulgaris* extract based on Gas chromatography-mass spectrometry (GC-MS) analysis.**

| No | Name | CAS Number | RT (min) | Area% | Quality |
|----|------|-----------|----------|-------|---------|
| 1 | Ethane, 1,1,2,2-tetraethoxy- | 003975-14-2 | 6.375 | 2.20 | 35 |
| 2 | 5-n-Propylindan | 092013-21-3 | 10.915 | 6.42 | 64 |
| 3 | 3-Acetylanisole | 000586-37-8 | 14.158 | 4.33 | 76 |
| 4 | (+) spathulenol | 077171-55-2 | 19.705 | 3.62 | 99 |
| 5 | 7,7-dichlorobicyclo[3.2.0]hept-2-en-6-one | 005307-99-3 | 22.2 | 3.13 | 58 |
| 6 | (-)-Loliolide | 005989-02-6 | 22.636 | 2.80 | 46 |
| 7 | 2-Pentadecanone, 6,10,14-trimethyl- | 000502-69-2 | 24.561 | 2.12 | 80 |
| 8 | NEOPHYTADIENE | 000000-00-0 | 24.67 | 1.27 | 98 |
| 9 | Methyl palmitate | 000112-39-0 | 25.89 | 2.49 | 98 |
| 10 | n-Hexadecanoic acid | 000057-10-3 | 26.611 | 22.97 | 99 |
| 11 | Methyl linoleate | 000112-63-0 | 28.442 | 1.19 | 99 |
| 12 | Methyl linolenate | 000301-00-8 | 28.505 | 3.16 | 98 |
| 13 | Phytol | 000150-86-7 | 28.868 | 16.69 | 91 |
| 14 | 9,12-Octadecadienoic acid (Z,Z)- | 000060-33-3 | 29.2 | 8.86 | 96 |
| 15 | 9,12-Octadecadienoic acid (Z,E)- | 000060-33-3 | 29.454 | 0.48 | 96 |
| 16 | Ethyl linolenate | 001191-41-9 | 29.522 | 1.33 | 95 |
| 17 | Glycerin 1,3-distearate | 002937-53-3 | 29.677 | 2.10 | 93 |
| 18 | Dipalmitin | 130548-52-6 | 34.176 | 5.15 | 78 |
| 19 | Eicosane | 000112-95-8 | 36.407 | 1.51 | 53 |
| 20 | Stigmasterol | 084122-10-1 | 43.105 | 2.18 | 50 |
| 21 | .gamma.-Sitosterol | 000083-47-6 | 43.998 | 3.54 | 68 |
| 22 | Oleic acid, 3-(octadecyloxy)propyl ester | 000000-00-0 | 45.129 | 1.83 | 50 |

validation serves as strong evidence for the efficacy of the employed methodologies in successfully identifying the top anticancer plants [41,42].

The confirmed anticancer plants have different range of anticancer metabolites. *Punica granatum* in addition to 61 anticancer metabolites, is rich of phenolic compounds including flavonoids, anthocyanins, catechins, complex phenolic, and hydrolysable tannin so has prevention effects on different type of cancers [43,44]. *Psidium guajava* with 26 anticancer metabolites and a comprehensive range of phenolic compounds has also a long history in medicinal usage and cancer treatment [45,46]. The next plants with 22 anticancer metabolites are *Salvia officinalis* and *Trifolium pratense* which have been used as anticancer plants and are rich in flavonoids and essential oils [47–50].

The other six anticancer plants that were listed in the Table 2 and S1 Fig, have different types of anticancer metabolites and their effects have been investigated widely [51–56]. Overall, this finding shows that plants exhibit diverse potential in the prevention and treatment of various types of cancer. Therefore, they deserve greater attention in therapeutic approaches.

In addition to experimentally confirmed anticancer plants, our results remarkably included nine novel previously undocumented anticancer plants (Table 3), based on literature reviews conducted during this study. Those plants which are from different families have various types of anticancer metabolites. *Diospyros kaki* is rich of phenolic and flavonoid components like hesperidin, quercetin, and luteolin which makes that a respected potential anticancer plant [57,58]. Another probable anticancer, *Angelica furcijuga*, has types of coumarins and phenylpropanoids components that strongly inhibit nitric oxide production (through nitric oxide synthase) as an effective factor in different cancers [59,60]. The remained plants including *Calligonum leucocladum*, *Juniperus thurifera*, *Ophiorrhiza liukiuensis*, *Sedum takesimense*, *Zanthoxylum dipetalum*, *Falcaria vulgaris*, and *Rhus wallichii* [61–64], have also had valuable antioxidant capacity. Previous studies have demonstrated that antioxidant activity is closely associated with the anticancer properties of plants [65–68]. These compounds possess aromatic and hydroxyl groups in their structure, which enable them to scavenge free radicals by donating unpaired electrons.

To validate the claim mentioned above, the hydro-alcoholic extract of *F. vulgaris* was tested on breast cancer cell lines. The results indicate an inhibitory effect of *F. vulgaris* extract on cell survival and growth, likely due to its strong antioxidant properties. The finding of the MTT assay indicate a clear concentration-dependent toxicity of *F. vulgaris* extract with highest lethality at the concentration of 30 mg/ml, highlighting the potent anticancer activity. Complementary evidences were provided by flowcytometry results that shown more than ninety percent cell death rate for both cell lines at concentration of 10 mg/ml of the extract. However, a different response was observed in higher concentrations between the cell lines, potentially indicating varying sensitivities or mechanism of resistance [69].

The main action of *F. vulgaris* on Breast cancer cell lines is apparently necrosis instead of apoptosis. This effect can be dependent on the high applied concentration of the extract (i.e., necrosis in the treatment of 30 mg/ml of the extract is more severe than that in 10 mg/ml), which probably leads to ROS-induced necrosis as an alternative anticancer strategy of plants metabolites [70–72]. Similar findings were recently reported that indicates cytotoxic effects of *F. vulgaris* extract on human liposarcoma cell line (SW-872) due to its terpenoid compounds [73]. The cerium oxide nanoparticles that were biosynthesized using *F. vulgaris* extract also showed anti-tumor effect on prostate cancer cell line (PC3) [74]. As the antioxidant capacity of the extract was stronger than known antioxidants like gallic acid and BHT, seems that the healing properties of *F. vulgaris* is depend on of its specific compounds and metabolites. Overall, the pronounced effects observed with *F. vulgaris* support its potential as a valuable candidate for developing natural anticancer therapies.

The anticancer properties of plants can often be attributed to their rich composition of phytochemicals such as fatty acids, phenolic, terpenoids and alkaloids compounds which can exhibit various mechanism of action like inducing apoptosis, inhibiting cell proliferation, antioxidant properties, anti-inflammatory effects and so on [75]. GC-MS analysis revealed that interestingly around one-fourth of the extract concentration (22.97%) is made up of the n-Hexadecenoic acid which has been extensively studied for its anticancer, anti-inflammatory, and antioxidant effects [76–78]. Literature review of the

molecular mechanisms of n-Hexadecanoic acid (palmitic acid) and its derivatives in tumor progression, demonstrating that palmitic acid affects various cancer cell lines through proliferation, apoptosis, metastasis, metabolic reprogramming, and autophagy [79]. It was also showed that palmitic acid increases oxidative stress through ROS generation, inhibiting RINm5F cell growth and promoting apoptosis [80]. As a major compound of the extract, n-Hexadecanoic acid demonstrated dose-dependent cytotoxicity in breast cancer cells via FAK/AKT pathway inhibition [81]. The n-hexadecanoic acid-enriched extract of *Halymenia durvillei* promotes apoptosis and autophagy of human triple-negative breast cancer cells [82].

Similarly, Phytol as a second high concentration compound (16.69%) in the *F. vulgaris* extract has been exanimated and showed anticancer properties possibly through necrosis and apoptosis mechanisms [83]. Phytol can down-regulate the expression of cancer stem cell markers (e.g., OCT4, NANOG, SOX2, ALDH1, ABCB1, CD44 and CD133) and reduce the proportion of side population cells in NCCIT human embryonic carcinoma cells [84]. Phytol's cytotoxicity against human lung carcinoma (A549) cells and the induction of bleb-like apoptotic bodies was also confirmed [85]. *Citrus unshiu* leaf extract containing phytol as a major compound induces autophagic cell death in human gastric adenocarcinoma AGS cells [86], and its synergistic effects with stigmasterol (another major component in *F. vulgaris*) enhance anticancer activity.

Additional compounds like Stigmasterol and spathulenol have emerged as noteworthy constituents [87–88]. Stigmasterol, as a phytosterol, exhibits anti-proliferative properties and induces apoptosis in the MCF-7 cell line, making it a promising antitumor compound by significantly decreasing of BCL-2 and BCL-XL genes expression [87]. Spathulenol, a combination of terpenes and antioxidants, demonstrates significant anti-proliferative, anti-inflammatory, and antimicrobial properties [73,89]. Another compound found in *F. vulgaris*, octadecanoic acid as anticancer and antioxidant metabolite like to n-hexadecenoic acid, demonstrates a remarkable ability to reduce the growth and migration of the SiHa cells probably through interaction with vascular endothelial growth factor [90]. In addition to the mentioned compounds which cover a significant percentage of the extract, there are other compounds like linoleate, oleic acid, and loliolide that their therapeutic effects have been investigated [91–93], and part of the anticancer properties of *F. vulgaris* can be dependent on these compounds.

While this study demonstrates the potent anticancer activity of *F. vulgaris* crude extract, we acknowledge that fractionation and isolation of its individual compounds remain essential for future research. The high abundance of bioactive metabolites like n-Hexadecanoic acid (22.97%) and Phytol (16.69%) in our GC-MS analysis, coupled with their established mechanisms, positions *F. vulgaris* as a promising candidate for drug discovery. This approach would bridge the gap between traditional ethnopharmacological use and modern precision oncology applications.

Considering the noticeable inhibitory effect of *F. vulgaris* extract on the growth of breast cancer cell lines, this finding reinforces our hypothesis that the methods utilized in this study have the potential to identify potential anticancer plants based on their metabolites. Moreover, this study represents an important preliminary step towards understanding the practical implications of harnessing the power of medicinal plants to inhibit the proliferation of breast or other types of cancer cells.

## Supporting information

**S1 Fig. The detailed information of top ten plants with the highest replication that their anticancer properties were previously established.** Venn diagram analysis showed the number of known anticancer metabolites in each top plant which were listed in the table of metabolites.
(DOCX)

**S2 Fig. The detailed information of nine potential anticancer plants that their anticancer properties were not previously established.** Venn diagram analysis showed the number of known anticancer metabolites in those plants.
(DOCX)

**S3 Fig. Gas chromatography – Mass spectrometry (GC-MS) analysis of *F. vulgaris* extract.**
(TIF)

**S4 Fig. Cytotoxicity of *F. vulgaris* extract on 3T3 normal fibroblast cells.** No significant cytotoxicity was observed at concentrations up to 30 mg/ml compared to the untreated control ($p > 0.05$).
(JPG)

**S1 Table. The list of 667 known anticancer metabolites.**
(XLSX)

**S2 Table. The list of sixty-one plant with cut off of ≥5 replications.**
(XLSX)

**S3 Table. Complete list of network analysis parameters used for plant ranking, including mathematical formulations and calculation algorithms.** The eight topological measures were computed using CytoHubba and Network Analyzer in Cytoscape 3.9.1.
(DOCX)

**S4 Table. Comparative study of IC50 values for different periods of incubation for the two cell lines (MCF7 and 4T1) exposed to different concentrations of *F. vulgaris* extract (5–90 mg/ml).**
(DOCX)

## Acknowledgments

The authors wish to thank S. Kazemi-noureini, M. Kheirabadi, for their excellent comments and review. We would like to express our sincere gratitude to the Department of Biology at Hakim Sabzevari University for their support and facilities provided during this research.

## Author contributions

**Conceptualization:** Eisa Kohan-Baghkheirati, Madjid Momeni-Moghaddam, Toktam Hajjar.

**Data curation:** Zahra Samadi, Zahra Ghavidel.

**Investigation:** Zahra Samadi, Zahra Ghavidel.

**Methodology:** Eisa Kohan-Baghkheirati.

**Project administration:** Eisa Kohan-Baghkheirati, Madjid Momeni-Moghaddam.

**Resources:** Eisa Kohan-Baghkheirati, Madjid Momeni-Moghaddam.

**Supervision:** Eisa Kohan-Baghkheirati, Madjid Momeni-Moghaddam.

**Validation:** Eisa Kohan-Baghkheirati, Madjid Momeni-Moghaddam, Toktam Hajjar.

**Writing – original draft:** Zahra Samadi, Zahra Ghavidel.

**Writing – review & editing:** Eisa Kohan-Baghkheirati, Madjid Momeni-Moghaddam, Toktam Hajjar.

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
