## [Decision Letter · Decision Letter 0]

27 Jun 2025

Thank you for submitting your manuscript to PLOS ONE. After careful consideration, we feel that it has merit but does not fully meet PLOS ONE’s publication criteria as it currently stands. Therefore, we invite you to submit a revised version of the manuscript that addresses the points raised during the review process.

We look forward to receiving your revised manuscript.

Kind regards,

Awatif Abid Al-Judaibi, PhD

Academic Editor

PLOS ONE

Journal Requirements:

Reviewers' comments:

Reviewer's Responses to Questions

**Comments to the Author**

1. Is the manuscript technically sound, and do the data support the conclusions?

Reviewer #1: Yes

Reviewer #2: Partly

Reviewer #3: Yes

2. Has the statistical analysis been performed appropriately and rigorously?

Reviewer #1: Yes

Reviewer #2: Yes

Reviewer #3: Yes

3. Have the authors made all data underlying the findings in their manuscript fully available?

Reviewer #1: Yes

Reviewer #2: Yes

Reviewer #3: Yes

4. Is the manuscript presented in an intelligible fashion and written in standard English?

Reviewer #1: Yes

Reviewer #2: Yes

Reviewer #3: Yes

Reviewer #1: This manuscript explores an integrative approach where network pharmacology and experimental validation are combined to search for novel anticancer botanicals, specifically focusing on Falcaria vulgaris (Sickleweed). The authors constructed a bipartite network of 3,250 plant species and 667 experimentally validated anticancer metabolites, resulting in the identification of 61 top-ranked plants. Among these, 85.25% had previously reported anticancer properties, and 14.75% were new candidates. For experimental validation, F. vulgaris was selected, showing significant cytotoxic and apoptotic effects on breast cancer cell lines (MCF-7 and 4T1). It also consisted of bioactive ingredients possessing known antioxidant, anticancer, and anti-inflammatory activities.

But there are some weaknesses Points:

The paper requires detailed explanation of network analysis:

Criteria for selection of metabolites and construct networks.

Algorithms and parameters used to rank plants.

Validation metrics for network predictions (e.g. sensitivity, specificity, cross-validation).

Why did you select Falcaria vulgaris for experimental validation over the other novel candidates identified in your analysis?

Did you test the cytotoxicity of F. vulgaris extract on any normal (non-cancerous) cell lines to assess selectivity and potential toxicity?

The highest concentrations used in cytotoxicity assays (up to 10 mg/mL) are rather high; discussion about physiological extensibility and possible toxicity in vivo is warranted.

Twenty-two compounds were obtained by GC-MS analysis, but there are some questions that the manuscript should meet:

What specific compounds the anticancer effect bring-about.

If the active compounds were fractionated or isolated.

If possible, comparison with known concentrations of these compounds in other anti-cancer studies.

Reviewer #2: Summary:

The authors identified potential anticancer botanicals using a network analysis from metabolites of thousands of medicinal plants. They shortlisted several candidates including F. vulgaris. They evaluated the potential anticancer effect of F. vulgaris extract on 2 breast cancer cell lines. The network analysis and metabolite information can be useful in discovering new plant-based cancer therapies. However, it is not clear how it can prevent cell growth and whether the effect observed is only in cancer cell lines and not in normal cells. Overall writing style and paragraph structure should be improved.

Major points:

- Overall concentrations of extract used are very high, more control experiments are needed to verify if the cell viability and anti-cancer activity is not just toxicity due to high concentration of extract. Please clarify whether the negative control used is the solvent used for the extract.

- Include which statistical analysis was performed for each figure in the respective figure legend.

- Figure 3: Not clear what control is, just cells with no compounds, a negative control of whatever solvent was used for the extract should be included. Also presenting the data in a dose response curve format would be better, showing concentration in the x axis. Also, it would be useful that once data is presented in a dose curve format the IC50 value is calculated.

- Figure 4: Again, it is not clear what control means, control should be the same amount of the extract solvent used.

- It would be nice to include in the introduction why extracts from plants are better than synthetic drugs used for cancer therapies, specifically targeted therapies. Are there any examples in the literature that show that plant extracts have less side effects than targeted therapies? Please elaborate on that.

- It is probably best to state that this is an exploration of F. vulgaris as a therapeutic candidate and not validation, as the exact mechanism of how is inhibits cell proliferation is not elucidated.

Minor points:

- Line 46. What is the evidence that metabolites don’t cause side effects on humans: Cite source and give example or remove sentence.

- Line 61. “Otherwise….” Simplify sentence for better readability.

- Line 62: “graph” features. Please be clear, what does graph features mean?

- Line 68: Why did they select Breast Cancer to test the anti-cancer properties, why not other cancers? Please elaborate on this on the intro. What kind of therapies are being used currently for Breast cancer.

- Figure 1: A, B and C: Please provide a legend for the color and size of each data point on the figure not only in the legend. D: Please explain better what “Replicate” means in the figure legend.

- Line 194: There is no figure 1e, please fix.

- Line 180: What is the relevance of that sentence in that paragraph.

- Line 232: Please explain better how the IC50 was calculated, it is not clear from the bar graph.

- Line 308: Change “methods” for approaches.

- Line 309: Fix grammar of sentence.

- It would be interesting to know from the metabolites found in the extract of F. vulgaris, which is the one that exhibits the major anti-cancer effect and if there is anything known in literature of how it prevents cell proliferation. Is it by inhibiting a specific oncogenic protein or is it affecting a cell growth pathway?

Reviewer #3: The authors provide a clear and concise description of the problem they intended to solve and the objective of their study in the introduction. The methods employed are also clearly stated and align with the results obtained. The manuscript is also presented in clear and concise English language.

**Do you want your identity to be public for this peer review?** For information about this choice, including consent withdrawal, please see our Privacy Policy

Reviewer #1: **Yes:** Dr. Abdulkarim Dakah

Reviewer #2: No

Reviewer #3: **Yes:** Moses Yeboah Addo

---

## [Author Response · Author response to Decision Letter 1]

10 Aug 2025

B. Reviewer #1: This manuscript explores an integrative approach where network pharmacology and experimental validation are combined to search for novel anticancer botanicals, specifically focusing on Falcaria vulgaris (Sickleweed). The authors constructed a bipartite network of 3,250 plant species and 667 experimentally validated anticancer metabolites, resulting in the identification of 61 top-ranked plants. Among these, 85.25% had previously reported anticancer properties, and 14.75% were new candidates. For experimental validation, F. vulgaris was selected, showing significant cytotoxic and apoptotic effects on breast cancer cell lines (MCF-7 and 4T1). It also consisted of bioactive ingredients possessing known antioxidant, anticancer, and anti-inflammatory activities.

Author Response: We sincerely thank the reviewer for their thoughtful evaluation and constructive comments on our manuscript. Their insightful observations regarding the network pharmacology approach and experimental validation of Falcaria vulgaris will certainly help improve the quality and clarity of our work. We have carefully addressed all points raised, as the reviewer's feedback has been invaluable in refining this study.

But there are some weaknesses Points:

1. The paper requires detailed explanation of network analysis:

-Criteria for selection of metabolites and construct networks,

Author Response: We appreciate the reviewer's valuable suggestion to elaborate on our network methodology. In response, we have expanded Section "Anticancer Metabolite List" to provide greater transparency about our approach (Page 6, lines 116-126). Specifically, we now detail: (1) the systematic database searches (PubChem, KNApSAcK) using validated keywords ("anticancer", "anti-tumor", "cytotoxic"); (2) our three-tier inclusion criteria requiring peer-reviewed experimental evidence, confirmed plant origin, and documented mechanisms of action; and (3) the construction of an unweighted bipartite network in Cytoscape 3.9.1 linking the 667 validated metabolites to their 3,250 host plants. The complete metabolite list is provided in Supplementary Table 1. While our focus was on compiling experimentally verified relationships rather than computational validation, we acknowledge this as an interesting direction for future methodological development. These clarifications strengthen the reproducibility of our study while maintaining its biological focus.

-Algorithms and parameters used to rank plants, validation metrics for network predictions (e.g., sensitivity, specificity, cross-validation).

Author Response: We sincerely appreciate the reviewer's valuable suggestions for improving methodological transparency. We have substantially revised Section “Network Analysis” to provide: (1) full details of the eight topological parameters used for plant ranking (Degree, Betweenness, Closeness, Edge percolated component, Eccentricity, Radiality, Stress, and Bottleneck scores calculated via CytoHubba/Network Analyzer), (2) a three-tier validation approach combining prospective experimental testing of F. vulgaris (selected when no anticancer reports existed) with retrospective literature validation to concordance with known anticancer species, and (3) explicit discussion of methodological considerations including our unweighted network approach. These additions (Page 6, Lines 131-143; Supplementary Table S3) provide complete transparency about our ranking algorithm while demonstrating its predictive value through successful experimental confirmation of F. vulgaris's anticancer properties.

2. Why did you select Falcaria vulgaris for experimental validation over the other novel candidates identified in your analysis?

Author Response: We selected Falcaria vulgaris for experimental validation based on the following considerations:

1. Novelty with supporting evidence: while our network analysis identified nine new potential anticancer plants (including Diospyros kaki, Angelica furcijuga, and Juniperus thurifera), F. vulgaris stood out due to:

Page 5, line 105-107; Traditional Use: documented therapeutic properties in ethnomedicine (antioxidant, antimicrobial, anti-inflammatory) [#21, #22]

Page 5, line 107-110; phytochemical richness: high abundance of bioactive compounds with known anticancer potential (e.g., flavonoids, caryophyllene oxide, spathulenol) [#23–#26]

Page 5, line 110-112; GC-MS Validation: confirmed presence of key anticancer-associated components (carvacrol, α-pinene) [#26]

2. Strategic Prioritization:

Page 5, line 112-114; Research Gap: Despite its phytochemical profile, F. vulgaris had limited direct evidence for anticancer effects, making it an ideal candidate for experimental validation.

Regional Relevance: As a native Iranian species, it aligned with our goal to explore understudied medicinal plants with local availability for future translational work.

3. Comparative Advantages: stronger preliminary data (vs. other novel candidates) from traditional use and phytochemistry, as well as higher feasibility for laboratory validation (extraction yields, compound stability)

We acknowledge that other eight candidates (e.g., Sedum takesimense) also show promise and merit future investigation (page 23, line 413-425). This rationale clarifies our selection process.

3. Did you test the cytotoxicity of F. vulgaris extract on any normal (non-cancerous) cell lines to assess selectivity and potential toxicity?

Author Response: We appreciate this important question regarding selectivity and potential toxicity. In addition to our cancer cell line testing, we evaluated F. vulgaris extract cytotoxicity in normal 3T3 fibroblast cells (Swiss albino mouse embryo origin) as a preliminary assessment of therapeutic window. The extract showed no significant growth inhibition (p>0.05) in normal cells at concentrations up to 30 mg/mL (MTT assay, 24 h exposure), while demonstrating potent, dose-dependent cytotoxicity in cancer cell lines (more than 90% cell death in 30 mg/mL). This preliminary selectivity suggests cancer-specific activity, though we acknowledge that further testing in additional normal human cell lines would strengthen these findings.

4. The highest concentrations used in cytotoxicity assays (up to 10 mg/mL) are rather high; discussion about physiological extensibility and possible toxicity in vivo is warranted.

Author Response: We appreciate this insightful observation regarding our extract concentrations. While 10 mg/mL exceeds typical plasma levels, this aligns with in vivo safety data from Goorani et al. (2019) showing F. vulgaris tolerability at 200 mg/kg in rats, Basatinya et al. (2021) reporting efficacy of 500 mg/kg ethanolic extract of F. vulgaris for 14 days in rats, and Jafari et al. (2022) reporting comparable in vitro doses (22.5 mg/mL). The higher concentrations were necessary due to crude extract limitations (only 22.97% active palmitic acid; reduced bioavailability) and because lower doses (0.15-2.5 mg/mL) showed no cytotoxicity in MTT/flow cytometry assays. Preliminary mouse studies (1500 mg/kg, unpublished) further support safety. While future pharmacokinetic studies will refine therapeutic windows, our selected doses (≥5 mg/mL) successfully established proof-of-concept activity against MCF-7/4T1 cells, as detailed in Results (Page 18, lines 338-347).

5. Twenty-two compounds were obtained by GC-MS analysis, but there are some questions that the manuscript should meet:

-What specific compounds the anticancer effect bring-about. If the active compounds were fractionated or isolated.

Author Response: We thank the reviewer for prompting this important clarification. Our GC-MS analysis identified 22 compounds in F. vulgaris extract, with two dominant classes collectively representing ~80% of the composition and demonstrating validated anticancer mechanisms: (1) Fatty acids/steroids (49.56%, e.g., n-Hexadecanoic acid and Stigmasterol), and (2) Terpenes (30.2%, e.g., Phytol and Spathulenol). The two most abundant bioactive compounds, n-Hexadecanoic acid (22.97%) and Phytol (16.69%), which collectively representing ~40% of the extract are strongly associated with the observed bioactivity. n-Hexadecanoic acid demonstrated dose-dependent cytotoxicity in breast cancer cells via FAK/AKT pathway inhibition (Deepti et al. 2024). The n-hexadecanoic acid-enriched extract of Halymenia durvillei promotes apoptosis and autophagy of human triple-negative breast cancer cells (Sangpairoj et al. 2022). Citrus unshiu leaf extract containing phytol as a major compound induces autophagic cell death in human gastric adenocarcinoma AGS cells (Song et al 2015), and its synergistic effects with stigmasterol (another major component in F. vulgaris) enhance anticancer activity. While these two components are primary candidates for the observed effects, we acknowledge that minor constituents (e.g., spathulenol, stigmasterol) may contribute to the extract's overall efficacy through potential synergistic interactions.

While we have not yet fractionated individual compounds (a limitation noted in page 27 line 493-495), the extract’s efficacy aligns with reported bioactive concentrations (page 25-27, line 455-492).

-If possible, comparison with known concentrations of these compounds in other anti-cancer studies.

Author Response: Thank you for this suggestion. Our comparison of key F. vulgaris compounds with concentrations reported in other anticancer studies shows similar trends. For example:

A. n-Hexadecanoic acid (palmitic acid) constituted 22.97% of our extract, equivalent to ~2.3 mg/mL (9 mM) at 10 mg/mL crude extract

B. Phytol represented 16.69% of our extract, equaling ~1.7 mg/mL (5.7 mM) at 10 mg/mL crude extract

Wang et al. (2023) reviewed the molecular mechanisms of palmitic acid and its derivatives in tumor progression, demonstrating that palmitic acid affects various cancer cell lines through proliferation, apoptosis, metastasis, metabolic reprogramming, and autophagy across concentrations ranging from 5 μM to 50 mM. The dose closest to our study (10 mM) was shown by Beeharry et al. (2004) to increase oxidative stress through ROS generation, inhibiting RINm5F cell growth and promoting apoptosis.

Regarding phytol, a literature review (Islam et al., 2023) reports an effective treatment range of 3-400 μM. However, some studies required extended treatment durations (e.g., 40 μM for 7 days; Soltanian et al., 2020). Other investigations, such as Thakor et al. (2017), used higher concentrations (exceeding those in our study) and demonstrated phytol's cytotoxicity against human lung carcinoma (A549) cells, with an IC50 of 16.97 ± 2.31 mM, and observed the induction of bleb-like apoptotic bodies.

Overall, while the concentration range applied in our study is relatively high, we selected doses ≥5 mg/mL of crude extract for the MTT and flowcytometry assays because lower concentrations (0.15, 0.31, 0.62, 1.25, and 2.5 mg/mL crude extract) did not show significant inhibitory effects on MCF-7 and 4T1 cell lines compared to the control (0 mg/mL extract).

C. Reviewer #2:

The authors identified potential anticancer botanicals using a network analysis from metabolites of thousands of medicinal plants. They shortlisted several candidates including F. vulgaris. They evaluated the potential anticancer effect of F. vulgaris extract on 2 breast cancer cell lines. The network analysis and metabolite information can be useful in discovering new plant-based cancer therapies. However, it is not clear how it can prevent cell growth and whether the effect observed is only in cancer cell lines and not in normal cells. Overall writing style and paragraph structure should be improved.

Author Response: We sincerely appreciate the reviewer’s valuable feedback, which has helped strengthen our manuscript. In response to the all major and minor comments, specially we have: (1) clarified the mechanistic basis of F. vulgaris metabolites by detailing their predicted interactions with key cancer pathways (apoptosis regulators, cancer stem cell markers, Bcl-2, cell cycle proteins CDK4/6 and cell cycle regulation), supported by additional references; (2) included and highlighted our preliminary cytotoxicity examination on normal 3T3 cells; (3) improved figure legends for better clarity; (4) enhanced the literature review regarding plant-derived anticancer agents; and (5) improved writing style. We hope these revisions and cross-referenced responses adequately address all concerns and have improved both the scientific rigor and readability of our work.

Major points:

1- Overall concentrations of extract used are very high, more control experiments are needed to verify if the cell viability and anti-cancer activity is not just toxicity due to high concentration of extract. Please clarify whether the negative control used is the solvent used for the extract.

Author Response: We appreciate the opportunity to clarify these critical methodological points. The higher concentrations (≥5 mg/mL) were necessary because lower doses (0.15-2.5 mg/mL) showed no cytotoxicity in MTT/flow cytometry assays and due to crude extract limitations (only 22.97% active palmitic acid; reduced bioavailability). While future pharmacokinetic studies will refine therapeutic windows, our selected doses (≥5 mg/mL) successfully established proof-of-concept activity against MCF-7/4T1 cells, as detailed in Results (Page 18, lines 338-347). On the other hand, these concentrations align with in vivo safety data from Goorani et al. (2019) showing F. vulgaris tolerability at 200 mg/kg in rats, Basatinya et al. (2021) reporting efficacy of 500 mg/kg ethanolic extract of F. vulgaris for 14 days in rats, and Jafari et al. (2022) reporting comparable in vitro doses (22.5 mg/mL).

Regarding to the solvent used for the extract, we should clarify that the hydroalcoholic extract (70% ethanol) was completely dried and reconstituted in sterile ddH₂O for treatments (page 7, line 148-153), ensuring No residual ethanol, and identical solvent conditions between treatment and control groups (sterile water as negative control).

2- Include which statistical analysis was performed for each figure in the respective figure legend.

Author Response: Thank you for this suggestion. We have now explicitly stated the statistical methods in each figure legend as follows: Figure 2 (Antioxidant assay), “Data is represented as Mean ± SD, n = 3.”; Figure 3 (MTT assay), “Data analyzed by one-way ANOVA with Tukey test (n=4 replicates; **p<0.0001 vs control)”; and Figure 4 (Flow cytometry assay), “Numbers in each quadrant reflects the percentage of cells (population).”.

Analyses were performed using GraphPad Prism 8 with α=0.05. These updates appear in the revised manuscript figure legends.

3- Figure 3: Not clear what control is, just cells with no compounds, a negative control of whatever solvent was used for the extract should be included. Also presenting the data in a dose response curve format would be better, showing concentration in the x axis. Also, it would be useful that once data is presented in a dose curve format the IC50 value is calculated.

Author Response: We appreciate this opportunity to clarify our controls. In Figure 3 (MTT assay): the "Control" (0 mg/mL extract) represents cells treated with the identical solvent used for the extract (sterile ddH₂O), same volume as in treated groups, and identical incubation conditions. To clarify this, we changed "0 mg/ml extract (C, as control)" to "0 mg/ml extract (C, solvent control, same volume ddH₂O as in treated groups)" in figure legend. Also, mentioned details were added to the Methods (italic texts):

Page 10, line 202-205: … the cells were treated with different concentrations of extract of F. vulgaris (0 mg/ml as solvent control contains same volume ddH2O with in treated groups, 0.15, 0.31, 0.62, 1.25, 2.5, 5, 10, 20, 30, 40, 50, 60, 70, 80 and 90 mg/ml) for 24, 48, and 72 hours.

Regard to presenting the data in a dose response curve format and calculating IC50 value, we appreciate the suggestion to present cytotoxicity data as dose-response curves. While we agree this format effectively illustrates IC50 calculations, we have retained the column charts as the current column format (Figure 3) directly compares effects across multiple con

---

## [Decision Letter · Decision Letter 1]

4 Sep 2025

Dear Dr. Kohan-Baghkheirati,

Thank you for submitting your manuscript to PLOS ONE. After careful consideration, we feel that it has merit but does not fully meet PLOS ONE’s publication criteria as it currently stands. Therefore, we invite you to submit a revised version of the manuscript that addresses the points raised during the review process.

We look forward to receiving your revised manuscript.

Kind regards,

Awatif Abid Al-Judaibi, PhD

Academic Editor

PLOS ONE

Journal Requirements:

**Additional Editor Comments:**

*F. vulgaris*
*F. vulgaris*

Reviewers' comments:

Reviewer's Responses to Questions

**Comments to the Author**

Reviewer #1: All comments have been addressed

Reviewer #2: All comments have been addressed

2. Is the manuscript technically sound, and do the data support the conclusions?

Reviewer #1: Yes

Reviewer #2: Yes

3. Has the statistical analysis been performed appropriately and rigorously?

Reviewer #1: Yes

Reviewer #2: Yes

4. Have the authors made all data underlying the findings in their manuscript fully available?

Reviewer #1: Yes

Reviewer #2: Yes

5. Is the manuscript presented in an intelligible fashion and written in standard English?

Reviewer #1: Yes

Reviewer #2: Yes

Reviewer #1: The author took all the required notes and answered some important questions I had, and the article is now ready for publication.

Reviewer #2: The authors did a great job in addressing the feedback. The readably was improved and scientific accuracy was clarified.

**Do you want your identity to be public for this peer review?** For information about this choice, including consent withdrawal, please see our Privacy Policy

Reviewer #1: **Yes:** Abdulkarim Dakah

Reviewer #2: No

---

## [Author Response · Author response to Decision Letter 2]

9 Sep 2025

Manuscript PONE-D-25-27242 R1

Response to Reviewers

Dear Prof. Dr. Awatif Abid Al-Judaibi,

Thank you for the opportunity to revise our manuscript “Network Pharmacology Approach Identifies Novel Anticancer Botanicals: Experimental exploration of Falcaria vulgaris (Sickleweed) as a therapeutic Candidate’’ for publication in PLOS ONE. We are grateful to you and the reviewers for the insightful comments and constructive feedback, which have significantly strengthened our manuscript. We have carefully addressed all points raised in the latest round of review. Specifically, we have incorporated the following key revisions as requested:

1. Reviewer-Recommended Citations: We confirm that we have reviewed all reviewer suggestions and have only added new citations where they were directly relevant to support new claims or clarifications made during revision, in accordance with journal policy.

2. Reference Integrity: We have performed a new, comprehensive verification of our reference list against PubMed and can confirm that all citations are correct, complete, and none have been retracted.

3. Normal Cell Testing & Dose Rationale: Detailed descriptions of the methodology for the 3T3 normal cell line assay and the rationale for the chosen high-dose range have been added to the Materials and Methods section. The corresponding results are now presented in the Results section and visualized in S4 Fig.

4. F. vulgaris Selection Rationale: We have now clearly integrated the multi-faceted rationale for selecting F. vulgaris over other candidate plants into the Result, detailing its traditional use, phytochemical richness, and the strategic research gap it addresses.

All changes made to the manuscript have been meticulously highlighted using the Track Changes feature for your convenience. Our detailed point-by-point responses to all comments are included in the section below.

We believe that these revisions have fully addressed the editorial and reviewers' concerns and have markedly improved the clarity, rigor, and overall quality of our work. We are confident that the manuscript now meets the high standards of PLOS ONE.

Thank you again for your time and consideration. We look forward to your final decision.

Sincerely,

Eisa Kohan-Baghkheirati

*S4 Fig. has been added.

** Reference number #1 has been edited in the revised version based on the source paper.

B. Additional Editor Comments:

Dear Authors,

Thank you for your submission to PLOS ONE. Based on the reviewers’ comments and your responses, please revise the manuscript to address the following:

1. Kindly, include details of the experiments performed to test the toxicity of F. vulgaris extract on non-cancerous cell lines, and provide the rationale for the chosen concentrations.

Author Response: We thank the Editor and reviewers (previous round) for this essential feedback. We have now integrated the requested details on the cytotoxicity assessment in normal cells and the rationale for dose selection directly into the Materials and Methods, and Results sections of the manuscript. The changes are as follows:

1. Materials and Methods (Page 10, lines 213-224): We have added details of specific normal cell line used (3T3 fibroblasts, Swiss albino mouse embryo origin), the culture conditions and assay used (MTT assay, 24h exposure), and the concentration range tested (0-90 mg/mL), aligning with the doses used on cancer cell lines. The rationale for using high concentrations, citing the crude nature of the extract, the low percentage of active compounds, and the precedent set by prior in vivo and in vitro studies on F. vulgaris (Ref. #22 and #23).

2. Results (Page 19, lines 375): We have added a new S4 Figure in the Results section presenting the findings from the normal cell assay, confirming the lack of significant cytotoxicity at concentrations up to 30 mg/ml, effective against cancer cells. This highlights the preliminary selective toxicity of the extract.

2. Clearly explain why F. vulgaris was selected over the other plants identified in the study. Please incorporate these points in the revised manuscript and resubmit for further consideration.

Author Response: We thank the Editor for this crucial feedback. We have now integrated a clear and detailed explanation for the selection of F. vulgaris for experimental validation directly into the Result section of the manuscript. This new paragraph outlines the multi-faceted rationale, balancing novelty with practical research considerations.

The new text (found on page 16, lines 316-333) explicitly states that F. vulgaris was selected based on:

1. Established Traditional Use & Bioactivity: Its documented ethnomedicinal properties (e.g., antioxidant, anti-inflammatory) provided a foundational safety and bioactivity profile.

2. Phytochemical Richness: GC-MS analysis confirmed a high abundance of metabolites with known anticancer associations (e.g., flavonoids, carvacrol, α-pinene).

3. Strategic Research Gap: It represented a high-potential but under-investigated candidate, allowing our study to generate novel, significant findings.

4. Feasibility and Translational Potential: As a regionally available species, it aligned with our long-term goal of exploring translatable therapeutic candidates.

5. We acknowledge the promise of the other highly-ranked plants from our analysis and have highlighted them as compelling candidates for future research.

C. Reviewers Comments: -

Author Response: All reviewer comments have been fully addressed in previous round. There are no new comments requiring a response at this stage.

We wish to express our sincere gratitude to Reviewers #1 and #2 for their time and effort in re-evaluating our manuscript. We are delighted that they found our revisions to be thorough and that the manuscript now meets the journal's standards for publication.

We also extend our thanks to the Editor for overseeing the review process. The constructive feedback provided throughout has been invaluable and has significantly strengthened the quality and clarity of our work.

---

## [Decision Letter · Decision Letter 2]

29 Sep 2025

Network Pharmacology Approach Identifies Novel Anticancer Botanicals: Experimental Exploration of *Falcaria vulgaris* (Sickleweed) as a Therapeutic Candidate

PONE-D-25-27242R2

Dear Dr. Eisa Kohan-Baghkheirati,

We’re pleased to inform you that your manuscript has been judged scientifically suitable for publication and will be formally accepted for publication once it meets all outstanding technical requirements.

Kind regards,

Awatif Abid Al-Judaibi, PhD

Academic Editor

PLOS ONE

Reviewers' comments:

Reviewer's Responses to Questions

**Comments to the Author**

Reviewer #1: All comments have been addressed

2. Is the manuscript technically sound, and do the data support the conclusions?

Reviewer #1: Yes

3. Has the statistical analysis been performed appropriately and rigorously?

Reviewer #1: Yes

4. Have the authors made all data underlying the findings in their manuscript fully available?

Reviewer #1: Yes

5. Is the manuscript presented in an intelligible fashion and written in standard English?

Reviewer #1: Yes

Reviewer #1: (No Response)

**Do you want your identity to be public for this peer review?** For information about this choice, including consent withdrawal, please see our Privacy Policy

Reviewer #1: **Yes:** Abdulkarim Dakah

---

## [Editor Report · Acceptance letter]

PONE-D-25-27242R2

PLOS ONE

Dear Dr. Kohan-Baghkheirati,

I'm pleased to inform you that your manuscript has been deemed suitable for publication in PLOS ONE. Congratulations! Your manuscript is now being handed over to our production team.

Kind regards,

on behalf of

Professor Awatif Abid Al-Judaibi

Academic Editor

PLOS ONE